# Diverse Prophage Elements of *Salmonella enterica* Serovars Show Potential Roles in Bacterial Pathogenicity

**DOI:** 10.3390/cells13060514

**Published:** 2024-03-14

**Authors:** Kirstie Andrews, Toby Landeryou, Thomas Sicheritz-Pontén, Janet Yakubu Nale

**Affiliations:** 1Centre for Epidemiology and Planetary Health, School of Veterinary Medicine, Scotland’s Rural College, Inverness IV2 5NA, UK; kirstieandrews2011@hotmail.co.uk (K.A.); toby.landeryou@sruc.ac.uk (T.L.); 2Center for Evolutionary Hologenomics, The Globe Institute, University of Copenhagen, 1353 Copenhagen, Denmark; thomassp@sund.ku.dk

**Keywords:** *Salmonella*, salmonellosis, prophage, serovars, virulence, zoonoses

## Abstract

Nontyphoidal salmonellosis is an important foodborne and zoonotic infection that causes significant global public health concern. Diverse serovars are multidrug-resistant and encode several virulence indicators; however, little is known on the role prophages play in driving these traits. Here, we extracted prophages from seventy-five *Salmonella* genomes which represent the fifteen important serovars in the United Kingdom. We analyzed the intact prophages for the presence of virulence genes and established their genomic relationships. We identified 615 prophages from the *Salmonella* strains, from which 195 prophages are intact, 332 are incomplete, while 88 are questionable. The average prophage carriage was found to be ‘extreme’ in *S.* Heidelberg, *S.* Inverness, and *S.* Newport (10.2–11.6 prophages/strain), ‘high’ in *S.* Infantis, *S.* Stanley, *S.* Typhimurium, and *S.* Virchow (8.2–9.0 prophages/strain), ‘moderate’ in *S.* Agona, *S.* Braenderup, *S.* Bovismorbificans, *S*. Choleraesuis, *S.* Dublin, and *S.* Java (6.0–7.8 prophages/strain), and ‘low’ in *S.* Javiana and *S.* Enteritidis (5.8 prophages/strain). Cumulatively, 61 virulence genes (1500 gene copies) were detected from representative intact prophages and linked to *Salmonella* delivery/secretion system (42.62%), adherence (32.7%), magnesium uptake (3.88%), regulation (5%), stress/survival (1.6%), toxins (10%), and antivirulence (1.6%). Diverse clusters were formed among the intact prophages and with bacteriophages of other enterobacteria, suggesting different lineages and associations. Our work provides a strong body of data to support the contributions diverse prophages make to the pathogenicity of *Salmonella*, including thirteen previously unexplored serovars.

## 1. Introduction

Nontyphoidal *Salmonella enterica* causes gastroenteritis, an important foodborne infection in humans which is transmitted via contaminated water or raw/undercooked food, especially poultry and pork [1,2]. Despite preventative measures through hygiene practices, pre-harvest intervention steps in animals, biosecurity, and surveillance strategies worldwide, salmonellosis rates remain high and a major cause of human morbidity and mortality [3]. Globally, *Salmonella* is estimated to cause up to 150 million infections and 155,000 deaths annually, the third most prominent cause of human deaths from diarrheal disease [4,5]. It is also the second most common cause of zoonotic disease in the EU, with an annual reported cases of up to 91,000, costing ~EUR 3b [6]. Similarly, *Salmonella* is one of the five main pathogens estimated to cause over 98% of deaths from foodborne illnesses, and results in over 10,248 infections and 33 deaths in the UK annually [7,8]. Within the UK, an estimated 800 salmonelloses cases are reported in Scotland each year, making this the second highest-reported cause of bacterial infectious intestinal disease in the region [9].

Over 2500 serovars of *S. enterica* have been reported, but only ~100 serovars are known to infect humans, with the majority of the strains being associated with increasing multidrug resistance to routine and frontline antibiotics, including colistin [10,11,12]. For example, monophasic *S.* Typhimurium strains are linked to microevolution of multidrug resistance and epidemiological success [13]. Furthermore, various virulence genes that are associated with cellular functionality (such as delivery/secretion systems, adherence, magnesium uptake, regulation, resistance to antimicrobial peptides, stress/survival, exotoxins, and antivirulence) are abundant in *Salmonella* genomes [14,15,16]. With declining antibiotic innovations, clearly the infection remains a serious public health concern, and thus must be urgently mitigated [12,17]. To effectively control salmonellosis, strategies must extend beyond developing alternative innovative control methods, but to also ascertain key factors that influence the distribution and spread of genetic features that underpin their diversity and evolution. These factors also impact *Salmonella* pathogenicity and drive the success of prevalent strains that are responsible for local and international infection outbreaks, and could direct treatment and control.

Bacteriophages (phages) are viruses which specifically infect bacteria and, like many bacteria, *Salmonella* interacts with phages in a plethora of ways, with concomitant impacts of either a lysis and release of amplified viral progeny by virulent phages, or integration into the host bacterial chromosome by temperate phages [18,19]. These integrated phages are called prophages, and replicate in a stable fashion with the host until they are spontaneously released or induced with mitomycin C, ultraviolet light, antibiotics, and hydrogen peroxide, or through nutrient starvation [19,20]. In general, prophages can exist as intact genetic elements that are functional and capable of inducing lysis. Alternatively, they can remain quiescent—being cryptic or incomplete—in which, through mutational degradation, have some of their genes repressed, and thus have lost the ability to effectively lyse their hosts [21,22]. Pertinent to *Salmonella enterica*, poly-lysogeny, the occurrence of abundant and diverse whole or degraded prophage elements, has been reported in their genomes, with an average of 5.29 prophages elements found in a single genome, which represents ~30% of the accessory genes in the strains [19].

Prophages can facilitate the transfer of antibiotic resistance genes and virulence factors which influence the ecological fitness, physiology, diversity, evolution, and pathogenicity of infected *Salmonella* strains [23,24]. This is mediated via horizontal gene transfer through several *Salmonella*–prophage and prophage–prophage interactions, thus promoting genetic diversity within the pathogen [19].

The *Salmonella*–phage interactions are mediated by either the host or prophage factors. The interactions driven by host factors involve phase variation (by Dam methylase with bacterial regulator, OxyR), silencing of exogenic DNA (by the H-NS protein from Hha family), transcriptional regulation (by the *invF* gene from AraC/Xy1S family and the *sopE* gene involved in the type three secretion system), and the two component system (the SsrAB of the *Salmonella* pathogenicity island 2, SPI-2) [19,25,26]. On the other hand, the interactions that are regulated by prophage factors can be seen in phage W104, which modifies the integration site *ryeA* in DT104 strain, leading to increased *ryeB* transcription and affecting the activity of related mRNA targets [19,27]. Also, the Gifsy1 phage activates the bacterial anti-terminator protein, AntQ, to form a stable complex with the host RNA polymerase, leading to DNA damage and death [19,28].

The *Salmonella* prophage–prophage interactions are mediated through cross-regulation (repressor/anti-repressor, as shown in Fles-2 and Gifsy 1 prophages), super-infection exclusion (controlled by SieA and SieB in P22 prophage), and repressor (C2 in P22 prophage) [19,29] genes. Other bacterial surface features regulated by prophages are related to glycosylation (GtrABC, SieA and repressor C2, the immunity factors of temperate phage, P22), O- acetylation (OafA, acyltransferase domain-related protein in SPC-P1 prophage), glycosidic bonds (phage beta polymerase) and adhesion (the *gpE* gene which codes for tail-spike protein of prophage found in LT2 strain) [19]. The carriage of *sodC* gene that plays a role in the establishment of *Salmonella* cells into macrophage contributes to virulence occurring through lysogenic conversion [30].

Historically, prophages play a significant role in *Salmonella*, as shown above, and remain a strong driving force to the diversity, pathogenicity, virulence, and evolution of the pathogen. However, studies conducted in the area were restricted to specific serovars/strains, with limited in-depth comparisons of the prophage repertoire of other prevalent *Salmonella* serovars and the potential role they play in salmonellosis [19,23,24,29,31,32,33,34]. To ascertain these impacts, we hypothesized that important *Salmonella* serovars carry diverse prophages which encode various virulence genes that potentially affect their bacterial host diversity and pathogenicity. Therefore, here, we went further to ascertain prophage carriage in fifteen prevalent non-typhoidal *Salmonella* serovars, including thirteen previously uncharacterized serovars in the UK and globally to complement these studies. To do this, we applied bioinformatics approaches to ascertain the carriage of diverse prophage elements in the genomes of five representative strains of the selected *Salmonella* serovars, and characterized the prophages present on the basis of their completeness. From the prophages identified here, we present a prophage classification system which was based on the total prophage load ascertained within the serovars and average prophage counts by the representative strains. Focusing on the intact prophages, we determined the virulence factors they encode, and linked these to *Salmonella* physiology and pathogenicity through genome functional analysis. Furthermore, we determined the genome diversity and associations formed between the extracted intact prophages across our dataset and with phages of other members of the *Enterobacteriaceae* to demonstrate their relationships and diversity.

## 2. Materials and Methods

### 2.1. Identification of Prevalent S. enterica Serovars, Collation of Genomes, and Extraction of Prophage Elements

We aimed to identify the virulence factors that are encoded in *Salmonella* prophages and their potential role in the host pathogenicity. To do this, we ensured that important serovars that are associated with the outbreaks of salmonellosis are targeted. First, we conducted a literature search to identify these serovars, which also included the ten most prevalent ones that are circulating within the UK and other parts of the world [35,36,37]. From the search, fifteen prevalent *Salmonella enterica* serovars were identified, and these include *S*. Enteriditis, *S.* Typhimurium, *S.* Newport, *S*. Infantis, *S.* Stanley, *S*. Agona, *S*. Java, *S*. Braenderup, *S*. Virchow, *S*. Javiana, *S.* Bovismorbificans, *S*. Dublin, *S*. Heidelberg, *S.* Choleraesuis, and *S*. Inverness (Table 1).

To enable a realistic wide coverage of the serovars for prophage analyses in this study, we downloaded five representative strains for each serovar (a total of 75 genomes were evaluated, Appendix A) from public databases for molecular typing and microbial genome diversity, PubMLST, https://pubmlst.org/ (Accessed on 6 February 2023), *Salmonella* species home in Enterobase, https://enterobase.warwick.ac.uk/species/index/senterica (Accessed on 9 February 2023) and genome/taxonomy browser in National Center for Biotechnology, NCBI, https://www.ncbi.nlm.nih.gov/ (Accessed on 9 February 2023). All strains examined in this study and their accession numbers are shown in Appendix A.

Each candidate *Salmonella* strain was subjected to genome analyses to identify and extract their prophage elements using PHASTER (PHAge Search Tool Enhanced Release) which performs BLAST (Basic Local Alignment Search Tool) against a customized prophage sequence database curated from NCBI [38,39]. PHASTER then clusters phage-like genes into prophage regions via DBSCAN (density-based spatial clustering of applications with noise), a data clustering algorithm [40]. Non-phage genes within these regions are then defined by a second BLAST search, which allows prophage regions to be given a completeness score reflecting the proportion of phage genes within the prophage region (intact, score > 90; questionable, score 70–90; and incomplete, score < 70) [38]. The genomes of the complete prophages identified in the bacterial strains were manually collated from PHASTER for later analyses.

### 2.2. Identification of Virulence Factors in Intact Prophages of Salmonella Strains

The genomes of the complete prophages were screened for the presence of *Salmonella* virulence genes using BLAST tool in the JavaScript-rich interface of the VFanalyzer tool within the Virulence Factor Database (VFDB) of bacterial pathogens, http://www.mgc.ac.cn/VFs/main.htm (Accessed on 9 February 2023). We searched the nucleotide sequences from the VFDB full (set B) database using the program BLASTn nucleotide query vs. nucleotide database. VFDB is a database of currently known bacterial virulent factors, and, within this, VFanalyzer can identify the virulence factors and antibiotic resistance genes encoded within the genomes [41]. VFanalyzer conducts intensive searches to identify sequence similarity in comparison to the pre-configured dataset within VFDB. VFDB provided an output summarizing the virulence factors found in each contig of the genome. The virulent factors were manually categorized into functions; delivery/secretion systems, adherence, magnesium uptake, regulation, stress/survival, exotoxins, and antivirulence within the pathogen (using *S.* Typhimurium LT-2 strain as a reference) [42]. The data was analyzed using GraphPad Prism 10 (GraphPad Software, Boston, MA USA).

### 2.3. Genome Diversity and Relationships of Salmonella Intact Prophages Examined Here with Phages from Other Salmonella Serovars and Enterobacteria

The intact prophage elements collected from PHASTER were further analyzed using PhageClouds (PhageClouds.ku.dk) [43]. The PhageClouds concept is a computational database which considers the whole genomes of phages in the system to create networks of interplay of gene/genomes of the phages. This concept has been successfully used to show distantly or closely related phages by the formation of clusters or clouds of prophages as shown in many bacterial strains [43]. The settings used were: distance threshold at ‘0.15’, source being ‘NCBI’ and host set as ‘*Salmonella*’ to ascertain relationships with other phages (and prophages) infecting and relating to various representative strains of the *Enterobacteriaceae* in the PhageClouds. Phages which do not share DNA do not form clouds [43].

## 3. Results

### 3.1. Diverse Prophage Elements Are Encoded in Salmonella Serovar Strains

In this study, we aimed to identify the repertoire of prophage elements that are encoded in the five representative genomes of the fifteen prevalent *Salmonella* serovars. We also aimed to further determine any virulence factors encoded in the intact viral genomes and their potential links to the pathogenicity of the pathogen.

From our genome analyses, diverse prophages were identified in the strains as shown in other studies [19,23,31,42]. A total of 615 prophage elements were extracted from the 75 *Salmonella* genomes we examined (Table 1). To provide a module for comparing the prophage populations across the *Salmonella* serovars and strains, we developed a classification method which is based on the cumulative prophage counts in each serovar and the average prophages carried by the representative strains in the serovars. Using this method, four (low, moderate, high, and extreme) categories of prophage carriage in *Salmonella* were identified (Table 1).

The first group is termed as the ‘low’ prophage carriage category. This group represents serovars that carried up to 29 prophages cumulatively in all the 5 representative genomes or an average of up to 5.8 prophages per strain within each serovar [19]. Thus, *S.* Javiana and *S*. Enteriditis serovars fall under the low prophage carriage category. The ‘moderate’ category encompasses serovars which encode a total of 30–39 prophages from the representative genomes in the serovars or an average of 6.0–7.9 prophages per strain within each of the serovars. Through this criterion, the *S.* Agona, *S.* Java, *S.* Bovismorbificans, *S.* Braenderup, *S.* Choleraesuis, and *S.* Dublin serovars are considered moderate prophage carriers. The *S.* Virchow, *S.* Stanley, *S.* Typhimurium, and *S.* Infantis, serovars encode a total of 40–49 prophages cumulatively in each of the serovars or an average of 8.0–9.0 prophages in the individual strains examined within the representative serovars. Hence, we considered the four latter serovars as the ‘high’ prophage carriage group. In the final category, the ‘extreme’ prophage carriage group, this represents genomes with a total of 50–59 prophages cumulatively in each of the serovars or greater than an average of 10 prophages per genome within the serovar. Thus, the *S.* Newport, *S.* Heidelberge, and *S.* Inverness serovars are regarded as the extreme prophage carriers (Table 1).

We observed that the cumulative prophage numbers carried in each examined serovar is generally commensurate with the average total prophage counts in the representative *Salmonella* genomes. However, there were still relatively much higher or lower numbers of prophages in the genomes of the representative strains than the average prophage counts in the serovars according to the classification method we used. For example, the *S.* Bovismorbificans serovar being considered a moderate prophage carrier serovar, yet we still observed one representative genome (Accession number ERR018000) which encoded only three prophages in its genome, as well as two other strains (Accession numbers SAMN09643854 and SAMN01924625) that encoded ten prophages each in their genomes (Appendix A). Conversely, the *S.* Javiana serovar, which we classified as a low prophage carrier, encoded an average of 5.8 prophages in the representative genomes. However, three strains (Accession numbers SAMN02345344, SAMN02335409 and SAMN02646184) within this serovar encoded 6–9 prophages in a single genome (Appendix A). Thus, the cumulative prophage carriage clearly concurred with the average prophage counts of strains within the serovars we examined. However, due to the wide differences in the range of prophage counts in each strain represented, we found that the model is more useful at representing the individual prophage carriage in each representative *Salmonella* genomes instead of the generalized carriage in the serovars.

Besides the diversity of prophage numbers encoded in the genomes of the strains representing the serovars, we also observed clear differences in the genome completeness of the prophages detected in the *Salmonella* strains (Figure 1, Table 1 and Appendix A) [23,44]. Generally, the numbers of complete (195) and incomplete (332) prophage regions detected across the dataset outweighed the number of questionable (88) prophage elements identified in the genomes examined [44]. Specifically, the *S.* Inverness strains showed the greatest number of intact (25) and incomplete (33) prophages, and *S.* Agona encodes the least intact prophages in their genomes (5 prophages in total, with a single intact prophage in each strain), while *S.* Javiana encodes the lowest number of questionable prophages (but only one prophage was each identified in 3 (Accession numbers SAMN02335409, SAMN02345148, SAMN02646184) of the 5 examined representative strains) (Appendix A). Interestingly, *S.* Agona and *S.* Inverness encoded no questionable prophages, but are among the highest incomplete prophage carriers (Appendix A). Only two strains lacked the presence of at least one intact phage region (*S.* Infantis strain SAMN06016056 and *S.* Javiana strain SAMN02335408).

### 3.2. Salmonella Prophages Encode a Large Number of Virulence Factors

After establishing the prophage diversity within the examined *Salmonella* serovars, we took further steps to ascertain the extent to which their genomes potentially contribute to the virulence of their hosts [42]. To do this, we investigated the prophage genomes exclusively to ensure that the virulence genes that were identified originated strictly from the prophages and related to *Salmonella* pathogenicity. We investigated all the intact prophages for this step since they have greater score (compared to the questionable or incomplete prophage), potentially functional hence, are the ideal candidates for this analysis as shown in other studies [23].

From the analyses of a total of 195 intact prophages using VFanalyzer, we identified 61 different virulence genes and a total of 1500 genes copies relating to various aspects of *Salmonella* virulence (Appendix A, Figure 2).

Most of the virulence genes, 26 genes (~42.6% of the genes) and 828 gene copies (55.2%% of the total gene copies) that were successfully identified from all the prophages examined are linked to *Salmonella* secretion systems [31,42]. This repertoire of genes is involved with the type III secretion system linked to *Salmonella* pathogenicity island 1 (SPI-1) which are responsible for epithelial cell invasion (Appendix A) [23,42]. We identified genes coding for invasin protein (*inv*) linked with *Yersinia pestis* which is the primary invasion factor in ex vivo tissue invasion models, and is necessary for efficient translocation of the bacteria across the intestinal epithelium in mice [45,46]. Other secretion effector protein genes (*sopE*, *sopE2*, *sseK2*, *pipB2*, *gogB*, *sspH2*, *sseI/srfH*, *sspH1*, *sifA*, *ospC4*, *aec27/clpV*, and *nleD-2*) for *Salmonella* and *Escherichia coli* (*nleB1*, *nleB3*, and *espO1-1*) were also identified [19,47]. The hypothetical prophage protein gene *ipaH* associated with the family of genes (*sspH1*, *sspH2*, and *slrP*) and the secretion effector found in P2-like and Gifsy *Salmonella* phages and is controlled by *mxiE* of *S. flexneri* 2a str.301 [48,49,50]. The *ompD* and *iroN* are outer membrane precursors, while *iroC*, *iroB*, *iroD*, and *iroE* are involved with the activity of salmochelin, a C-glucosylated enterobactin produced by *Salmonella* [51]. The *fadE29* and *msbB2* codes for Acyl-CoA dehydrogenase from *Mycobacterium avium* 104 [52]. The Lipid A biosynthesis lauroyl acyltransferase in *Salmonella* control the growth of the pathogen under carbon dioxide depletion [52,53]. The copies of these virulence genes are highly variable in the intact prophages, with the most prevalent ones being the *pipB2* and *sspH2* (with up to 14 copies found in all prophages of the serovar *S.* Agona and a single prophage of *S.* Newport, NewportD_8) (Figure 2, Appendix A). The secretion system genes are the only virulence determinants we found in *S.* Agona, and the carriage is consistent in all the prophages in this serovar (Figure 2, Appendix A).

The second highest set of virulence factors are linked to the adherence process with 20 genes (32.7% of the genes detected) and 454 gene copies (30.2% of the total gene copies found) from the prophages examined (Appendix A). The genes are related to the components of the host fimbriae *csgG, csgD, csgF, csgE, fimW, fimD, fimH, fimY, fimC, fimA, fimZ*, *fliD*, *fljA*, *fliC*, *fljB*, *stdA*, *stdD*, *mshB*, and *bapA* [19]. (Figure 2, Appendix A). At least a single copy of each of the adherence genes are found in all the prophages of the serovars except in *S.* Heidelberg, *S.* Typhimurium, *S.* Infantis, *S.* Choleraesuis, *S.* Agona, and *S.* Enteritidis which lacked any of the adherence genes in their intact prophages. Identical copies of *fljA*, *flic*, and *fljB* are found in prophages of *S.* Newport, *S.* Virchow, *S.* Braenderup, and *S.* Bovismorbificans (Figure 2).

In total, 2 genes, *mgtB* and *mgtC* (3.8% of genes identified), and a total of 28 gene copies (1.8% of total gene copies detected) are linked to functions related to nutrition, specifically for magnesium uptake, in the intact prophages of *S.* Java (Figure 2, Appendix A) [54].

Three regulatory genes, *phoQ*, *phoP*, and *rpoS* (4.9% of the genes identified, and 1.8% of the total gene copies), are found in *S.* Choleraesuis intact prophages only (Figure 2, Appendix A) [55].

Similarly, the functional involvement of the remaining virulence factors encoding for toxins, *sigA/rpoV*, *cnf*, *astA*, *pltA*, *pltB*, and *cdtB* (6 genes, 9.8% of the genes) and 27 copies (1.8% of gene copies), and the 2 immune modulation genes *gtrA* and *gtrB* (4% of the total genes), and 28 gene copies (1.8%) were detected in *S.* Heidelberg and *S.* Inverness, *S*. Typimurium, *S*. Dublin, *S.* Choleraesuis, and *S.* Java. The *sodCI* gene (0.47% of the genes in the prophages) which encodes for stress, survival, and replication within macrophages had 83 gene copies (5.5%), and is encoded by all the serovars examined, except the *S.* Infantis and *S*. Stanley strains (Figure 2, Appendix A) [19,56,57].

### 3.3. Genomes Diversity of Intact Prophages Identified in This Study and Their Relationships to Phages of Other Members of Enterobacteria

We have established the diversity of all prophages we extracted from the examined *Salmonella* strains and ascertained the contribution the representative intact prophages make to the pathogenicity of their hosts. In this section, we went further to determine the genome diversity of our intact prophages and their relationships with other *Enterobacteriaceae* including other *Salmonella* strains in the PhageClouds database [43,58].

From our analysis, we observed diversity in the sizes of our prophage genomes (indicated as yellow dots, Figure 3) compared to other *Salmonella* phages [21,42,59]. Our phage genomes are considerably smaller than *Salmonella* SPFM phages in cloud A, phages related to SKML-39 and SFP10 (infecting both a *Salmonella* and *E. coli*) in cloud B, and phage Munch infecting S. Newport, 7t3, and SE_PL in cloud C [60,61,62]. Conversely, some of our phage genomes are much more similar or larger in size to phages found in clouds E, F, and G, which are related to phages such as *Salmonella* phage vB_Sen_STGO-35-1, S134, and BP63, respectively (Figure 3) [63,64].

We identified seven clouds (clusters of related phages, clouds 1–7) formed by our phages (Figure 3). The largest cloud, cloud 1, contains 61 genomes of the 195 intact prophage genomes we extracted, and encompasses prophages from *S.* Dublin, *S.* Bovismorbificans, *S.* Javiana, *S.* Newport, *S.* Virchow, *S.* Choleraesuis, *S.* Typhimurium, *S.* Enteritidis, and *S.* Inverness (Figure 3). The phage DNA in cloud 1 shared similarities to, and clustered with, Gifsy_1 and Gyfsy_2 phages [23]. Clouds 2 and 3 contain 26 and 22 prophage genomes, respectively. In cloud 2, our phages from *S.* Agona, *S*. Enteriditis, *S*. Braenderup, *S*. Newport, *S.* Javiana, *S.* Java, *S.* Bovismorbificans, *S*. Dublin, and *S.* Stanley share similarities to *Klebsiella* phages ST11 and ST437-OXA245, and *Salmonella* phage Fels2 (Figure 3) [65,66]. Our phages in cloud 3 are prophages from *S.* Bovismorbificans, *S*. Enteriditis, *S*. Dublin, *S.* Inverness, and *S.* Choleraesuis, and share DNA identity to *Salmonella* phage_ST64B (Figure 3) [67]. Four of the *S*. Choleraesuis prophages in cloud 3 are located around the boundaries of the cluster and distantly related to cloud 1 phages (Figure 3). Cloud 5 contains 21 prophages from *S*. Infantis, *S.* Heidelberg, *S*. Virchow, and *S*. Typhimurium, and shares similarities to *Salmonella* phages such as ST-39 and *Shigella* phage_Sf6 [63,68]. Cloud 6 has 17 prophages from *S*. Braenderup, *S.* Javiana, *S*. Inverness, and *S*. Stanley, but are mostly akin to *E. coli* phages P2 and *Yersinia* virus_L413C, and share similarities to phages in clouds 5 [69,70]. The final cloud, cloud 7, contains only 1 of our phages from *S.* Newport, and was found to be identical to *E. coli* phage_ESS12 and_BSP161 and *Salmonella* phages vB_SalM-LPST153 and SalM-LPST144 (Figure 3) [71].

## 4. Discussion

The AMR crisis in salmonellosis continues to plague the application of medicine in human and agricultural healthcare systems. Innovative strategies are being developed to ensure that the treatment of salmonellosis remains effective in population medicine [72]. Whilst novel preventive strategies are important and can contribute to solving this health crisis, there are still unmet needs to ascertain the various factors that drive the virulence of specific *Salmonella* serovars that are responsible for human and animal infections [19]. Advancements in this area would greatly complement current and prospective treatment strategies, identify vaccine targets, and direct surveillance mechanisms by providing molecular markers for *Salmonella* infection as well as other bacterial zoonoses.

Lytic phages provide an invaluable tool for the control of *Salmonella* infection, and the strategy of using whole or derived phage products is an attractive therapeutic approach either as a standalone treatment or as adjuncts to other anti-infectives such as antibiotics in different model systems [73,74]. On the other hand, temperate phages have limited therapeutic efficacy due to their potential to mediate horizontal gene transfer, induce lysogenic conversion, or transduce infected hosts [23,24]. However, temperate phages are hugely important at providing incredible insight into the pathogenicity of their hosts [19,21,23]. Thus, in this study, we applied bioinformatics approaches and established these impacts in *Salmonella* by identifying the presence of diverse temperate phages that are carried in representative genomes of fifteen important serovars. Then, we further studied the role of the phages in driving *Salmonella* virulence by establishing the presence of abundant related genes in the phage genomes and their links to *Salmonella* physiology and pathogenicity. Our strategy of using bioinformatics approaches for these analyses is well established, and concurs with methods used by previous researchers [23,42]. The increase in publicly available genomes, which is made possible due to better accessibility to sequencing platforms and high performing analyses tools, has greatly enhanced the efficiency of gene/genome predictions, and supports phenotypic observations [75,76].

To provide a broad perspective of how prophages contribute to *Salmonella* pathogenicity, we prioritized a wide range of the prevalent serovars that are responsible for both human and animal infections in the United Kingdom and include dominant serovars in many parts of the globe [35,36,37]. Of the fifteen serovars that were investigated, the *S.* Enteritidis and *S.* Typhimurium are among the leading serovars responsible for animal infection and zoonoses. It was reported that the two serovars alone contributed to a significant 48.7% of all human salmonelloses in England and Wales in 2012, although this has been reported to decrease to 47%, as seen from the 2022 data [77,78]. In Scotland alone, where the work was conducted, the two serovars are responsible for approximately 58% of the infections, as reported in 2020 [36]. In addition, cumulatively, the fifteen serovars examined contribute to ~62% and 74% of infections in England, Wales, and Scotland, respectively, and are reported to be resistant to multiple antibiotics, including colistin [10,11,12,79]. The serovars are also reported to be prevalent in various animal species which have a high potential for zoonoses or transmission. For example, the *S.* Typhimurium contributes to ~46.7 infection in pigs, and *S.* Dublin is the leading serovar responsible for infection in cattle, with 61.6% of total infections recorded in the animal species in 2022 [78].

The occurrence of diverse prophage carriage in *Salmonella* strains has been reported in *S.* Typhimurium, *S.* Brandenburg, *S.* I:4,5,12:i:-, *S.* Reading, *S.* Rissen, *S.* Kentucky, *S.* Enteritidis, and *S.* Gallinarium isolates infecting humans and animals [19,23,42]. Our observation of the diverse prophages in all the 75 genomes from the fifteen serovars we examined concurs with these earlier reports. Our data also provide the broadest study ever conducted in the area, with data on thirteen additional serovars which were not studied previously [19,23,42]. Consequently, our data provide a significant extended body of evidence to support the frequent occurrence of poly-lysogeny in *Salmonella* serovars, which has also been reported in typhoidal *Salmonella* strains, as well as other *Enterobacteriaceae* such as *Clostridioides difficile* and *E. coli* [31,59].

We observed a higher number of defective prophages (incomplete and questionable prophages) than intact prophage elements in the genomes studied here, and this concurs with general previous observations in *Salmonella* strains irrespective of serovar [19,23,42]. This is also a common occurrence in many other bacterial pathogens such as *Paenibacillus larvae*, *C. difficile*, and *Bacillus thuringiensis* [44,59,80]. The majority of the strains we examined encoded ‘low’ to ‘moderate’ number of prophages, and this is comparable to several other reports on *Salmonella* strains which reported an average of 5.29 prophages in the pathogen [19]. Similarly, the number of intact prophages we observed concurs with the above reports as well. Conversely, however, we observed a cumulative average of 8.2 prophages, with many of the serovars and strains encoding a much higher number of prophages (‘high’ or ‘extreme’ prophage carriers) compared to observations above (Table 1 and Appendix A). These differences may be attributed to the method used in extracting the prophages in the previously mentioned studies, where the prophages were manually curated after PHASTER analyses and limited to certain genome thresholds, or focused on intact prophages only [23,42].

The vast gene determinants we observed in the prophages showed clear indications of their role in *Salmonella* virulence, and, consequently, to the host pathogenicity, physiology, and survival, and potentially to evolution [42]. Genes that are associated with secretion systems and adhesion constitute the highest population of the genes we observed, and are common in all the serovars that were examined, with few strain exceptions. The functions of secretion system genes are linked to SPI-1 and SPI-2, which facilitate *Salmonella* phagocytosis and internalization, protects the bacteria against anti-infectives and harsh conditions, and can lead to systemic infection. This clearly indicates the mutual relationships of the prophages and their hosts, and the carriage of these gene cargos may be highly essential for the bacterial existence [19,42]. Specifically, the *SopE* promotes pathogen entry into non-phagocytotic host cells (such as epithelial cells) by triggering actin cytoskeleton rearrangements, which stimulates membrane ruffling [81]. This gene was also found to play an active part in the formation of the early *Salmonella*-containing vacuole, which indicates that other prophage-encoded virulence factors may have multiple functions in addition to virulence [82]. The virulence genes identified here have been previously reported, albeit other serovars have their primary functions related to adhesion to surfaces, especially to gut lining and epithelial cells, and to abiotic surfaces to help establish infections and quorum sensing responsible for biofilm formation [19,21,83].

The regulatory proteins *phoQ* and *phoP* have been reported to activate the transcription of *pmrAB*, which encode a two-component regulatory system involved antimicrobial peptide resistance and lipid A modifications in *S.* Typhimurium carried by Gifsy-2 and Fels-1 prophages [55,84,85]. The anti-virulence factor, *grvA*, has also been described in Gifsy-2 phage, and this concurs with our observation on the strong association we saw with these phages in the phage clouds [20,33]. The gene works in tandem with *SodCI* to inhibit the virulence of strains which carry it, and strains which lack it were found to be less virulent [19]. Although interaction of the two genes is not clearly understood, there is an indication of their role in decreasing the pathogenicity of their hosts, and the output of prophage genes is influenced by interactions with pre-existing regulatory networks, as shown previously [33]. The advantage of possessing such seemingly counter-productive genes is not fully understood. However, it could be theorized that they are potential tools utilized by the bacteria in the pursuit of striking the fine balance of pathogenesis, such that the survival of the host is ensured to protect the bacteria’s own continued proliferation. As these genes are encoded by the prophages, which can be acquired or lost within a single recombination event, they may be useful in allowing rapid adaptation, and this supports our observation of the genes being located in other bacterial genomes as well as prophages of *S. flexneri*, *E. coli*, and *M. avium*, and potentially shows the exchange of genes among these organisms [23,48,52,68,86,87].

The typhoid toxin genes and other heat stable genes carried by the prophages shows their potential influence in the pathogenicity of the *S.* Choleraesuis, *S.* infantis, *S.* Newport, and *S.* Inverness. Even though these serovar are non-typhoidal, the carriage of the genes in some of the strains agrees with other bacteria, such as the carriage of the toxin in certain strains of *E. coli* [23,85,86]. These are extremely important virulence factors affecting the pathogenicity of the serovars studied, and this could further support their prevalence in human infections and in many animal host species. Since only the most prevalent serovars were analyzed for this part of the work, it could be expected that this gene is overrepresented in this cohort of serovars, and further work is required to verify this in other serovars.

Clearly, the high similarity we observed with the prophages extracted in this study suggested a strong interchange of genes among prophages of enteric bacteria, as shown by the relationships we observed in the various clouds formed with many gut bacteria. This also supports the various gene interactions and recombination events occurring among the prophages with surrounding bacteria in situ (humans and animals, and the environment) [66]. Also, previous work on *C. difficile* supports spontaneous inductions and lysogenisation within the human gut, and this could support the inter-change of genes among unrelated host bacteria, although more work is needed to support this in *Salmonella*. Phages lack a universal gene to study their diversity, as seen in bacterial 5, 16, and 23 S rRNA [59]. Although many genes such as capsid, holin, and portal proteins have shown some level of discrimination among phages, these have been shown to be extremely limiting, and some of the genes are not found in some phages [59]. The lack of universal conserved genes for phages makes this tool particularly useful in showing gene relationships, but further work is needed to identify other potential algorithms to cluster phages and enhance comparisons among them.

## 5. Conclusions

In this study, we conducted genome analyses of a total of 75 *Salmonella* strains from 15 prevalent serovars, including 13 serovars which were not analyzed in the UK and, indeed, globally. We showed that genetically diverse prophages are encoded in the strains, with the incomplete prophages being more prevalent than intact prophage elements in all the strains examined. Based on the average prophage counts in the serovars and strains, they were classified as either ‘low’, ‘moderate’, ‘high’, or ‘extreme’ prophage carriers. However, this classification method was found to be best at representing individual strains rather than the serovar due to the wide variability in the prophage load in each representative genome of the serovars examined. We observed various virulence genes from the 15 serovars we examined, including 13 serovars which were previously unexplored. This further supports that prophage carriage is universally present in *Salmonella*. The virulence factors identified in the prophages clearly indicate their links to secretion system, adherence, nutrition (magnesium uptake), stress, antivirulence, toxins, regulation, and resistance to antimicrobial peptides. These impacts may be mediated through horizontal gene transfer via various recombination events between *Salmonella*, their prophages, and phages of other close or distantly related enteric bacteria, as shown by the cloud relationships they formed. Further work would focus on prophage carriage in other uncharacterized serovars and strains and the role the prophages play in the identification, surveillance, and evolution of the *Salmonella* serovars.

## Figures and Tables

**Figure 1 cells-13-00514-f001:**
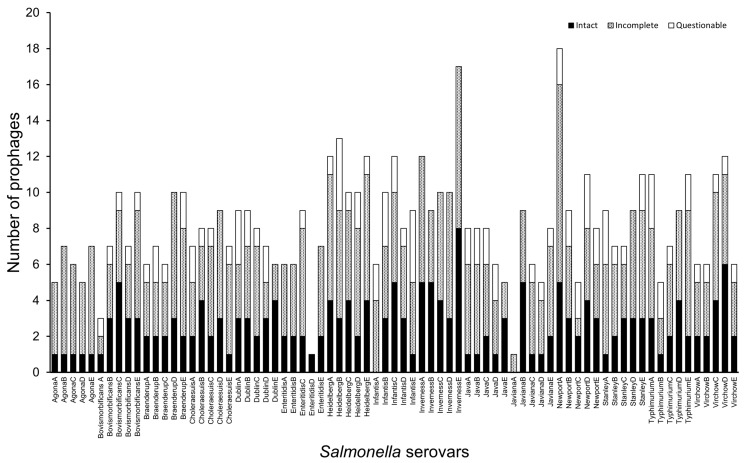
Prophage counts in seventy-five *Salmonella* genomes representing fifteen prevalent serovars in the UK and globally. The genomes were analysed using PHASTER and the intact, incomplete, or questional prohage numbers from each of the strains examined are shown. The suffixes A, B, C, D, and E in each serovar name represent the five strains examined for the serovar and the corresponding accession numbers of the strains are shown in Appendix A.

**Figure 2 cells-13-00514-f002:**
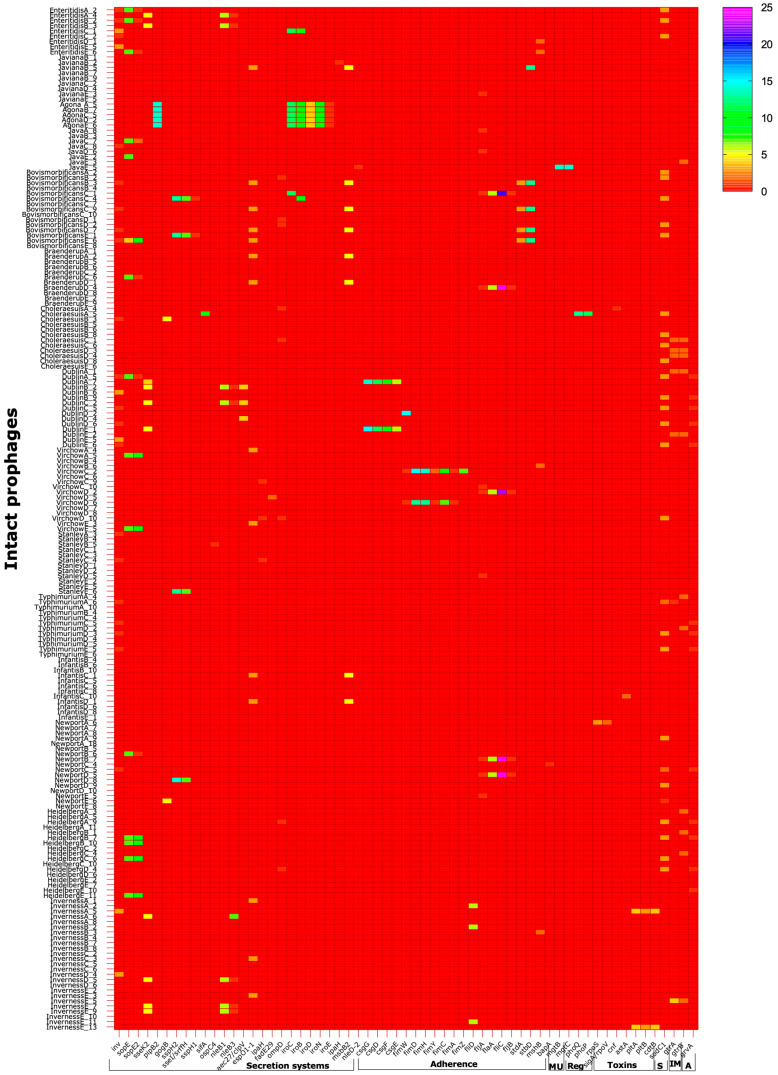
Heatmap showing the genes and gene copies of the 61 different virulence factors we identified in the genomes of all the 195 intact prophages examined in this study and assigned to functions: secretion systems, adherence, magnesium uptake (MU), regulation (Reg), toxins, stress and survival (S), immune regulation (IM), and antivirulence (A). The virulence genes were determined using the search option of the VFanalyzer tool in the virulence factor database of bacterial pathogens using the *S.* Typhimurium LT-2 genome as a reference and compiled in GraphPad Prism 10 (GraphPad Software, Boston, MA USA) to produce the heatmap.

**Figure 3 cells-13-00514-f003:**
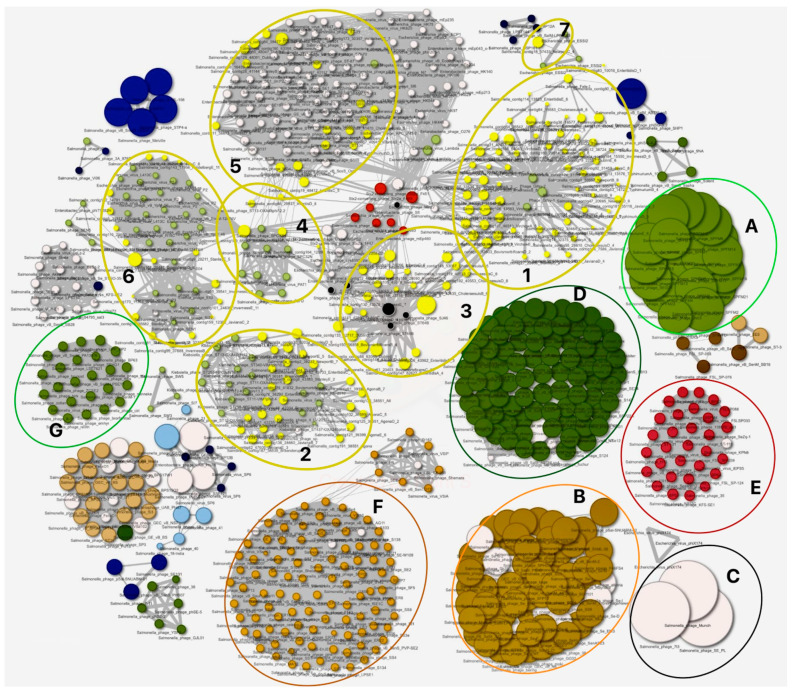
Genome diversity and relationships of the 195 intact prophages extracted from the 75 *Salmonella* genomes examined in this study with phages on the PhageClouds database. Each phage is represented by a single dot, and the size of each dot is related to the genome size, while the lines linking the genomes show where there are genome similarities and forming clusters or clouds. Clouds 1–7 show clusters formed by our prophages (yellow) and other phages in the database. Some of our phages have relatively smaller genomes compared to other phages in clouds A, B, and C, while others are much similar or larger in size compared to phages found in clouds D, E, F and G.

**Table 1 cells-13-00514-t001:** Prophage counts in the genomes of Salmonella serovars examined in this study as predicted using PHASTER.

Serovars	Total Prophage Elements					
	Intact	Incomplete	Questionable	Total	Average per Strain	Carriage Category
*S.* Enteritidis	9	19	1	29	5.8	Low
*S.* Javiana	9	17	3	29	5.8	Low
*S.* Agona	5	25	0	30	6	Moderate
*S.* Java	8	19	8	35	7	Moderate
*S.* Bovismorbificans	15	17	5	37	7.4	Moderate
*S.* Braenderup	11	22	6	39	7.8	Moderate
*S.* Choleraesuis	12	22	5	39	7.8	Moderate
*S.* Dublin	15	16	8	39	7.8	Moderate
*S.* Virchow	16	20	5	41	8.2	High
*S.* Stanley	12	24	7	43	8.6	High
*S.* Typhimurium	12	23	8	43	8.6	High
*S.* Infantis	12	21	12	45	9	High
*S.* Newport	17	23	11	51	10.2	Extreme
*S.* Heidelberg	17	31	9	57	11.4	Extreme
*S.* Inverness	25	33	0	58	11.6	Extreme
Totals:	195	332	88	615	8.2	High

The five representative genomes from the fifteen prevalent serovars examined in this study were downloaded from public databases and analyzed using PHASTER to identify the prophages encoded in them. All predicted phages were classified based on their completeness (intact, incomplete, and questionable), and the commutative counts in each serovar, and average carriage by the representative strains, are shown in the table. Details of phage carriage in individual strains are shown in Appendix A. The prophage counts are classified as ‘low’ (up to an average of 5.8 prophages/strain or cumulative of 29 prophages/serovar), ‘moderate’ (6.0–7.9 prophage/strain or 30–39 prophages/serovar), ‘high’ (8–9 prophages/strain or 40–49 prophages/serovar), and ‘extreme’ (more than 10 prophages/strain or 50–59 prophages/serovar).

## Data Availability

All strains used in this study are publicly available, and their accession numbers can be found in Appendix A.

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
