# Peer review of "Diverse Prophage Elements of Salmonella enterica Serovars Show Potential Roles in Bacterial Pathogenicity"

_cells, 2024, doi:10.3390/cells13060514_

Round 1

Reviewer 1 Report

Comments and Suggestions for Authors

The manuscript "Prophage elements of Salmonella enterica serovars show potential role in bacterial pathogenicity and diversity" by Andrews et al. described prophages in selected S. enterica serovars. The study demonstrated in detail the virulence genes occurring in prophage regions that may be important for bacteria fitness and pathogenicity. Therefore, this work is relevant to understanding the evolution, pathogenicity mechanisms and developing control strategies for this important pathogen.

General concept comments:

Lines 101-104: There are other extensive studies on Salmonella prophages than those cited in references 19 and 32, including the study cited in reference 23. Please include this reference and adjust the paragraph to accommodate it since it is a comprehensive study similar to the one conducted. I strongly suggest that the authors expose in the introduction what is new about their work and what complements previous work.

Line 126-127: The authors should better explain why they used "five representative strains for each serovar" instead of all available genomes. Since there is relatively great diversity in the occurrence and sequence of prophages even among closely related bacteria (this is true for your own data, in table S1, in the results lines 203-215 and so on), the strategy chosen by the authors should be justified.

Lines 132-133: Most accession numbers provided in Table S1 are from SRA, which is a sequence read database, not a genome database. Please use GenBank, RefSeq, or Assembly IDs in Table S1 and throughout the text to clarify which genomes were used and make the analysis reproducible.

Line 251-253: This argument is not strong enough. You could use other strategies to identify virulence genes in all intact prophages, not only using VFanalyzer (https://doi.org/10.1371/journal.pone.0239792, https://doi.org/10.1128/msphere.00452-21, https://doi.org/10.1038/s41598-022-25968-8). Besides, 195 intact prophage genomes are not too many. I think the effort of analyzing all of them would be worth it to have robust and comprehensive results. How can you be sure of the distribution of prophage virulence genes in Salmonella serovars if you haven't analyzed all the data you've collected?

Line 257: The title of this table does not reflect the data presented since not all intact prophages were analyzed — only those of few Salmonella serovars.

Line 258-260: What do you mean by "coverage"? I think a better term is "score". Better than citing these parameters (they have already been cited in the materials and methods) would be why intact prophages should be analyzed. For example, they are probably functional prophages and/or correctly predicted by the phaster, so they have a greater chance of conferring additional features to the host and/or the genes being transferred horizontally.

Specific comments:

Line 174: I don't understand why you cited other works if it was your results. Instead, cite a table/figure for these results, or use these references only in comparisons with your results.
Line 245-247: Please rewrite this sentence. If you are justifying an analysis based on the results of another paper (39), make that explicit in the sentence. For example, "Since prophages contribute to Salmonella genetic diversity and pathogenicity [39]..."
Line 539: Some DOI have links (https), but others do not. Please use them consistently. Also, many scientific names in the citations are not in italics.
Line 544: The reference 2 has a "1,2" after the article name. Please correct.
Lines 615 and 617: There is unsual line breaks in references 32 and 33. Please verify.

Author Response

Reviewer 1

General concept comments:

Lines 101-104: There are other extensive studies on Salmonella prophages than those cited in references 19 and 32, including the study cited in reference 23. Please include this reference and adjust the paragraph to accommodate it since it is a comprehensive study similar to the one conducted.

We have given detailed studies on the carriage of prophages in Salmonella and their role in the pathogenesis of the bacterium. Please see line 61-100.

For additional relevant references on work reported in the area we added six references. Please see line 106.

I strongly suggest that the authors expose in the introduction what is new about their work and what complements previous work.

Thank you for this suggestion. We have edited the last paragraph in the introduction to highlight work we have done and how our findings further research in the area. Please see lines 109-122

Line 126-127: The authors should better explain why they used "five representative strains for each serovar" instead of all available genomes. Since there is relatively great diversity in the occurrence and sequence of prophages even among closely related bacteria (this is true for your own data, in table S1, in the results lines 203-215 and so on), the strategy chosen by the authors should be justified.

As correctly indicated above, we have shown that the prophage distribution within each serovar is quite variable and this is a strong finding and not subject to change if we analyse all the genomes in the database.  There are ~40,000 thousand strains from the UK of the serovars we have studied in public databases. Based on our objectives, it is not feasible to examine prophage content of all the representative strains in the database, analyse the virulence genes in all the intact prophages to be extracted and discuss them in the detail as we did. Many other similar studies which have only studied representative strains show feasibility of doing these kinds of studies. We appreciate this is an excellent idea but far beyond the scope of this work, hence we would include this as future work. Please line 586-587.

Lines 132-133: Most accession numbers provided in Table S1 are from SRA, which is a sequence read database, not a genome database. Please use GenBank, RefSeq, or Assembly IDs in Table S1 and throughout the text to clarify which genomes were used and make the analysis reproducible.

We have made these corrections. Please updated Table S1 and lines 214-248

Line 251-253: This argument is not strong enough. You could use other strategies to identify virulence genes in all intact prophages, not only using VFanalyzer (https://doi.org/10.1371/journal.pone.0239792, https://doi.org/10.1128/msphere.00452-21, https://doi.org/10.1038/s41598-022-25968-8). Besides, 195 intact prophage genomes are not too many. I think the effort of analyzing all of them would be worth it to have robust and comprehensive results. How can you be sure of the distribution of prophage virulence genes in Salmonella serovars if you haven't analyzed all the data you've collected?

Thank you for this comment. We have analsysed the virulence genes from all the 195 intact prophages and edited Tables S2 and S3, Figures 2 and the whole   manuscript accordingly.

Line 257: The title of this table does not reflect the data presented since not all intact prophages were analyzed — only those of few Salmonella serovars.

The comment has been taken care of by the edits in the above comments.

Line 258-260: What do you mean by "coverage"? I think a better term is "score". Better than citing these parameters (they have already been cited in the materials and methods) would be why intact prophages should be analyzed. For example, they are probably functional prophages and/or correctly predicted by the phaster, so they have a greater chance of conferring additional features to the host and/or the genes being transferred horizontally.

We have changed the word ‘coverage’ to ‘score’ as advised and we agree this better reflects the information we are trying to convey. We also have removed the information on the completeness as we have already highlighted this in the methods as rightly spotted by reviewer 1. See lines 275-278We have highlighted intact prophages being functional in lines 68-68.

Reviewer 2 Report

Comments and Suggestions for Authors

Dear Authors,

Although I completely agree that the topic of your manuscript is very relevant, unfortunately I do not find it acceptable for publication in the current form. Your title "Prophage elements of Salmonella enterica serovars show potential role in bacterial pathogenicity and diversity" is not being reflected in the body of the manuscript, more precisely in the most relevant parts: methods and results. To be specific, you state that you applied bioinformatics approaches to ascertain the prophage elements of representative strains of important Salmonella serovars however you admittedly say that for this purpose you used only well established methods previously used by researchers. This is not a problem in itself, however, not going one step further is an issue. To provide some examples, where this additional step would be beneficial: 1. Instead of relying on an arbitrary criterion (the classification method/model as it is called in the manuscript) you should have invest more effort into assessing the underlying distributions of data (prophage genomic load) and draw your conclusion based on understanding of this distribution. Some additional clustering algorithms would also be strongly advised for this task. You did notice that variance is quite great in each group and your text does not justify individual serovar placement within groups accordingly which is a great objection that should be handled. Another key flaw of the current version of the manuscript is the last paragraph in results section: Genomes comparisons of intact prophages identified. This section is not complete and it does not reflect the title. In the text which follows you simply used a clustering tool to group the prophages identified, but more is needed in order to justify the "genome comparison" part of the title. My strong suggestion is to focus on these points and then resubmit the manuscript because the topic is of interest and with these major issues covered, I believe it would make a good publication.

Regards

Author Response

Reviewer 2

Although I completely agree that the topic of your manuscript is very relevant, unfortunately I do not find it acceptable for publication in the current form.

Your title "Prophage elements of Salmonella enterica serovars show potential role in bacterial pathogenicity and diversity" is not being reflected in the body of the manuscript, more precisely in the most relevant parts: methods and results.

To be specific, you state that you applied bioinformatics approaches to ascertain the prophage elements of representative strains of important Salmonella serovars however you admittedly say that for this purpose you used only well established methods previously used by researchers.

This is not a problem in itself, however, not going one step further is an issue.

To provide some examples, where this additional step would be beneficial:

Instead of relying on an arbitrary criterion (the classification method/model as it is called in the manuscript) you should have invest more effort into assessing the underlying distributions of data (prophage genomic load) and draw your conclusion based on understanding of this distribution. Some additional clustering algorithms would also be strongly advised for this task.

You did notice that variance is quite great in each group and your text does not justify individual serovar placement within groups accordingly which is a great objection that should be handled.

This is an excellent idea however it is beyond the aim of our manuscript. The aim of our work is to determine the variability in the different phage elements found in Salmonella, and focus on the intact prohages to determine their potential role in salmonellosis. Considering the prophage genomic load in each strain we examined would not provide us with the opportunity to detect the intact prophages to carry out these objectives. We understand this may provide an insight for any clustering among the serovars hence have included as future work. See lines 568-570. We have removed the word diversity in the title of manuscript to avoid further confusion associated with it and clustering.

Another key flaw of the current version of the manuscript is the last paragraph in results section: Genomes comparisons relationship of intact prophages identified. This section is not complete and it does not reflect the title. In the text which follows you simply used a clustering tool to group the prophages identified, but more is needed in order to justify the "genome comparison" part of the title. My strong suggestion is to focus on these points and then resubmit the manuscript because the topic is of interest and with these major issues covered, I believe it would make a good publication.

We agree with this suggestion. Since there are no universal genes for phages and the confusion came from the title of the sub-section, we have decided to revise the title of the sub-section to reflect our findings. See lines 381-382.

Round 2

Reviewer 1 Report

Comments and Suggestions for Authors

The authors have addressed all the requirements indicated.

A piece of advice I would give for future work would be to use an identifier that directs to the fasta sequence of the bacterial genome, for example, GCA_000006945.2 (Genbank) instead of SAMN02604315 (BioSample) or PRJNA57799 (BioProject). This doesn't compromise the study, but it does make it difficult for other authors to use the exact data for reproducibility or to complement the results in the future.

All in all, the work is good.